# Hybrid Re-matching for Continual Learning with Parameter-efficient Tuning

**Weicheng Wang**[1], **Guoli Jia**[3], **Xialei Liu**[1], **Liang Lin**[4], **Jufeng Yang**[1,2,5*]

[1] VCIP & TMCC & DISSec, College of Computer Science, Nankai University, Tianjin, China.
[2] Pengcheng Laboratory, Shenzhen, China.
[3] Department of Electronic Engineering, Tsinghua University, Beijing, China.
[4] School of Computer Science and Engineering, Sun Yat-sen University, Guangzhou, China.
[5] Nankai International Advanced Research Institute (SHENZHEN·FUTIAN), Shenzhen, China.
`2120230639@mail.nankai.edu.cn, exped1230@gmail.com`
`xialei@nankai.edu.cn, linliang@ieee.org, yangjufeng@nankai.edu.cn`

## Abstract

Continual learning seeks to enable a model to assimilate knowledge from non-stationary data streams without catastrophic forgetting. Recently, methods based on Parameter-Efficient Tuning (PET) have achieved superior performance without even storing any historical exemplars, which train much fewer specific parameters for each task upon a frozen pre-trained model, and tailored parameters are retrieved to guide predictions during inference. However, reliance solely on pre-trained features for parameter matching exacerbates the inconsistency between the training and inference phases, thereby constraining the overall performance. To address this issue, we propose HRM-PET, which makes full use of the richer downstream knowledge inherently contained in the trained parameters. Specifically, we introduce a hybrid re-matching mechanism, which benefits from the initial predicted distribution to facilitate the parameter selections. The direct re-matching addresses misclassified samples identified with correct task identity in prediction, despite incorrect initial matching. Moreover, the confidence-based re-matching is specifically designed to handle other more challenging mismatched samples that cannot be calibrated by the former. Besides, to acquire task-invariant knowledge for better matching, we integrate a cross-task instance relationship distillation module into the PET-based method. Extensive experiments conducted on four datasets under five pre-trained settings demonstrate that HRM-PET performs favorably against the state-of-the-art methods. **The code is available in the** `https://github.com/wei-cheng777/HRM-PET`.

## 1 Introduction

Faced with the non-stationary data flow in the real world, *continual learning* (CL) [83, 56, 65] aims to overcome *catastrophic forgetting* [76, 97, 31, 101] while acquiring knowledge from a sequence of tasks [57, 49, 69]. CL is crucial for achieving artificial general intelligence [39, 82, 84], garnering significant attention and advancements in recent years [38, 107, 52, 92, 70]. In this paper, we focus on the challenging class incremental learning (CIL) scenario [90, 34, 18, 102].

Rehearsal-based method is one of the most representative approach [32, 79, 8], which stores a portion of historical examples in a rehearsal buffer to alleviate catastrophic forgetting [5, 54, 50]. Consequently, the information of old classes is accessible when training the new classes. However, this

---

*Corresponding author.

39th Conference on Neural Information Processing Systems (NeurIPS 2025).

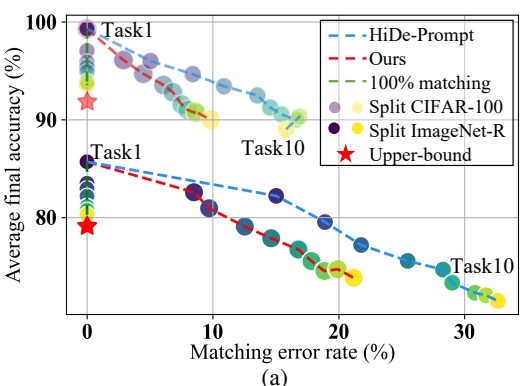
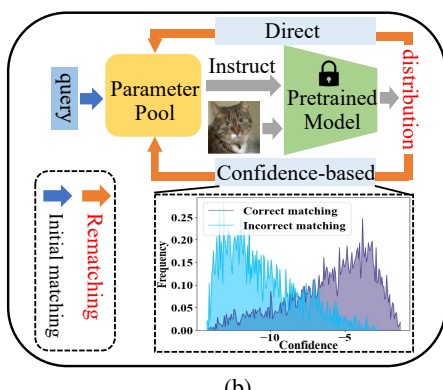

Figure 1: (a) shows comparisons of matching and average final classification accuracy of our method with HiDe-Prompt on CIFAR-100 and ImageNet-R across ten tasks with Sup-21K. (b) presents that during HRM-PET inference, the predicted distribution after the PET is leveraged for re-matching.

strategy raises privacy concerns [71, 72], and the performance is sensitive to storage capacity [88, 43]. Hence, *rehearsal-free* continual learning [89, 23, 3, 47] receives increasing attention.

Rehearsal-free methods train models without the help of rehearsal buffer [105, 55, 46, 21]. Among them, Parameter-Efficient Tuning (PET) [73, 95, 35] maintains the powerful feature extraction ability of pre-trained models, thereby shows impressive performance. PET updates a few parameters when adapting to downstream tasks, and frozes the pre-trained model to alleviate the catastrophic forgetting [58, 20, 61]. Most PET-based methods [88, 80] maintain a parameter pool, then train isolated parameters for each task. Particularly, the corresponding group of parameters in the pool is distinguished by the task identity [77]. During inference, the task identity of samples is unknown. Current methods address this issue by matching appropriate task identity for testing [88, 80]. DualPrompt [88] employs the QKV mechanism [78] to match task-specific prompts. LGCL [36] introduces language guidance during the matching. HiDe-Prompt [80] enhances the inference of task identity through the approximated distributions derived from uninstructed representations.

Although many works make efforts to improve the parameter matching process, there still exists a substantial potential for further enhancement. As shown in Fig. 1a, over 15% of data from Split CIFAR-100 are mismatched by the sota method HiDe-Prompt. Furthermore, when testing on the Split ImageNet-R, the matching accuracy is only 67.58%. As a result, the classification accuracy is 4.54% and 8.85% lower than the setting with 100% matching accuracy. To sum up, previous methods determine the task identity through a one-shot matching, which performs sub-optimal in CIL setting. Consequently, we propose a re-matching mechanism to calibrate the matching results.

The re-matching mechanism is shown in Fig. 1b. Specifically, we propose a Hybrid Re-Matching Parameter-Efficient Tuning (HRM-PET) method, which contains two re-matching operations. First, owing to the shared knowledge in the parameter pool, the task identity of the predicted class for some samples may be correct, despite the incorrect initial matching. These samples can be detected when the predicted class does not belong to the matching task identity. The task identity is directly replaced with the one obtained from the prediction. Second, we design confidence-based re-matching for all samples with incorrect initial matching. Specifically, for the parameters and the class of data mismatched, they have never been trained together. Therefore, the confidence of these predictions may be lower than the correct matching, and the experimental results in Fig. 1b demonstrated this assumption. To this end, we detect incorrect matching based on the confidence of the predicted distribution, and obtain the corrected task identity from the class with the sub-highest score. Moreover, we aspire to directly re-match more samples in the first scenarios. Therefore, we integrate cross-task instance relationship distillation [7] into the PET-based methods, which encourages aligning the shared feature space from different task parameters and improves the task-invariant knowledge.

In summary, our contributions are as follows: 1) We propose a hybrid re-matching strategy based on predicted distribution to improve matching and classification accuracy in PET-based CL. 2)We further introduce cross-task instance relationship distillation to align the shared feature space, which

promotes the learning of task-invariant knowledge. 3) We empirically validate the effectiveness of HRM-PET on extensive datasets using five distinct pre-trained models.

## 2    Related Work

### 2.1    Continual Learning

Many methods have been proposed for CL [40, 63, 99, 93, 12], such as regularization-based [2, 1, 85], rehearsal-based [9, 24, 11] and architecture-based methods[94, 106, 30]. Recently, PET-based methods [37, 73, 60, 98] show strong performance in rehearsal-free setting, which leverage parameter-efficient tuning techniques [33, 44], such as prompt tuning, to adapt a sequence of tasks upon frozen pre-trained models. L2P [89] employs the QKV mechanism [78] to select several prompts corresponding to each sample in a shared prompt pool. In addition to task-specific prompts, DualPrompt [88] incorporates global prompts to acquire task-invariant knowledge. LGCL [36] introduces language guidance at the task and instance level to improve prompt matching. HiDe-Prompt [80, 81] decomposes the PET-based method into three components, simultaneously enhancing each part, and has achieved state-of-the-art performance. Instead of selecting parameters, CODA-prompt [73] transforms the prompt selection process into a weighted mechanism. DAP [35] and APG [75] propose to generate prompts. Moreover, LAE [19] utilizes ensembling [48] to propose a unified CL framework for three Parameter-Efficient Tuning. Although MOS [74] also addresses mismatching through a self-refined retrieval mechanism, the assumptions are different from ours: MOS assumes that the sample is mismatched when the task identity of the final prediction is not the same as the initial matching. In contrast, HRM-PET detects and corrects mismatching based on confidence rather than only task identity consistency.

### 2.2    Parameter-Efficient Tuning

Parameter-Efficient Tuning [10] is designed to tune pre-trained models with fewer parameters while freezing most pre-trained parameters. For example, Adapter [28, 64] inserts small network modules between the layers of a pre-trained model. Prompt Tuning [66] adds lightweight trainable parameters (prompts) to the input of the pre-trained model. Compared with Prompt Tuning, Prefix Tuning [44] maintains the length of the output sequence unchanged. LoRA [29] introduces low-rank matrices that capture task-specific adaptations. VPT [33] applies prompt tuning to ViT for the first time, designing learnable input prompts that activate the pre-existing knowledge within a pre-trained model. The capability of Parameter-Efficient Tuning to acquire task-specific knowledge renders it highly conducive to continual learning [16, 17, 96].

## 3    Methodology

### 3.1    Continual Learning Formulation

Continual Learning aims to learn a sequence of tasks without forgetting. In CIL, each task is consisting of $n_t$ samples $D_t = \{(x_1^t, y_1^t), (x_2^t, y_2^t), \ldots, (x_{n_t}^t, y_{n_t}^t)\}$ from a set of several classes, where $t$ is task identity, $x_i^t \in \mathcal{X}$ represents the $i$-th input image and $y_i^t \in \mathcal{Y}$ is corresponding class label. Therefore, a total of $T$ tasks are defined as $D = \{D_1, D_2, \ldots, D_T\}$. As shown in Fig. 2, at time $t$, only $D_t$ of task $t$ is available for training. For inference, a unified linear classifier $g_{\theta_g}$ must distinguish all classes encountered so far. Note that the classes between tasks will not overlap and task identity is unknown during inference following common CIL settings[57, 67, 38]. The deep model $f_\theta : \mathcal{X} \to \mathcal{Y}$ parameterized by weights $\theta = \{\theta_h, \theta_g\}$ is often considered as incremental learners, which is split as a feature extractor $h_{\theta_h}$ and linear classifier $g_{\theta_g}$. In PET-based methods, lightweight parameters $p$ combined with a frozen pre-trained model $h_{ptm}$ is trained as $\theta_h$ to extract image features. For input test image $x$ from arbitrary tasks, the model outputs probability distributions $g(h(x; \theta_h); \theta_g)$ to predict its label $\hat{y}$. For rehearsal-based CIL, when training the model on $D_t$, a small number of exemplars from $\{D_1, \ldots, D_{t-1}\}$ are available in the storage buffer. This paper deals with the more challenging rehearsal-free setting, where images in the past are never stored.

We follow HiDe-Prompt [80] as our baseline. Given the image $x \in D_t$ of current task $t$, the frozen pre-trained model $h_{ptm}$ with parameters $\theta_{ptm}$ is directly used to extract its features as query

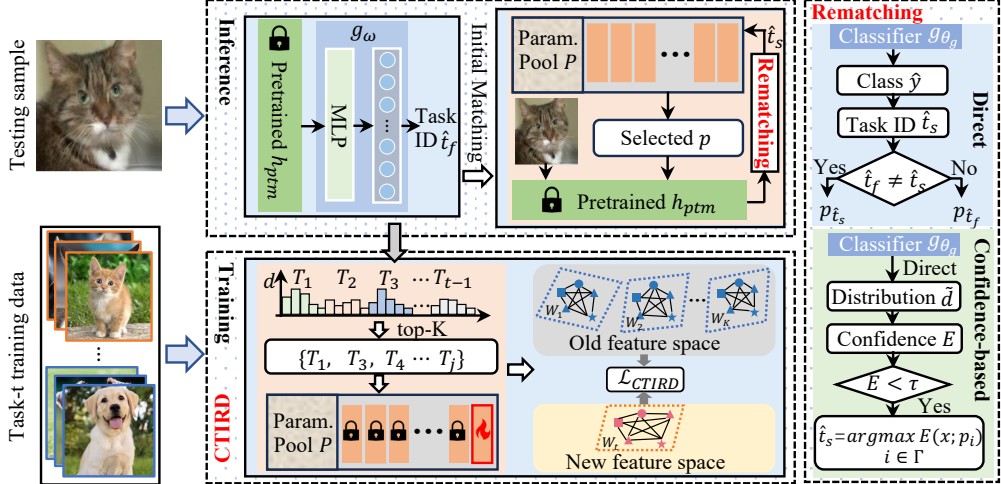

Figure 2: Pipeline of our method. Task ID means task identity. $\hat{t}_f$ and $\hat{t}_s$ denote the outcomes of the initial matching and the re-matching. $p_{\hat{t}_f}$ and $p_{\hat{t}_s}$ represent the corresponding parameters of $\hat{t}_f$ and $\hat{t}_s$ in the parameter pool. Re-matching consists of two methods: Direct and Confidence-based scenarios. $\Gamma$ is a set of top-N predicted task identities from initial matching.

$q(x) = h(x; \theta_{ptm})$. Then, the query feature is input to an auxiliary classifier $g_\omega$ parameterized by $\theta_\omega$ for the prediction of task identity, $\hat{d}(x) = g(q(x); \theta_\omega)$ represents the output class-level probability distribution, which is optimized by cross-entropy during the training. A parameter (LoRA) pool $P = \{p_0, p_1, \dots, p_t\}$ is maintained. During training, we get probability distribution $\tilde{d}(x) = g(h(x; p_t, \theta_{ptm}); \theta_g)$ by using the pre-trained model followed with current parameter $p_t$. Then, $p_t$ of the current task $t$ and classifier $g_{\theta_g}$ are optimized by cross-entropy. To maintain valid prediction distributions for past tasks, we follow HiDe-Prompt to model the categorical features as a Gaussian distribution. For further details, please refer to the Supplementary Materials.

## 3.2 Hybrid Re-matching

### 3.2.1 Direct Re-matching

During the inference phase as shown in Fig. 2, previous PET-based methods first employ feature query $q(x)$ to predict the task identity [88, 80]. The corresponding parameter is matched from $P$ and attached to the pre-trained model, then predicts the final classification result $\hat{y}$. When the initial matching is incorrect, the $\hat{y}$ is likely to be confused. However, for some simple samples, the features extracted by shared information in the parameter pool can also be distinguished. Therefore, the task identity of predicted label $\hat{y}$ for these samples may be correct. Replacing the initial matched task identity with this can readily enhance matching accuracy. Therefore, We propose a simple *direct re-matching* (DRM) solution by mapping the predicted class-level distribution $\tilde{d}(x)$ obtained with the initial matched parameters back to the task identity and conducting a parameter re-matching strategy:

$$
\begin{aligned}
\hat{t}_f &= \mathcal{T}(\underset{i}{\arg\max}\, g(q(x); \theta_\omega)), \\
\hat{t}_s &= \mathcal{T}(\underset{i}{\arg\max}\, g(h(x; p_{\hat{t}_f}, \theta_{ptm}); \theta_g)),
\end{aligned}
\tag{1}
$$

where $\mathcal{T}$ converts class into task identity, $\hat{t}_f$ and $\hat{t}_s$ represent the outcomes of the initial matching and re-matching, respectively. Undoubtedly, we cannot confirm the correctness of $\hat{t}_s$. To cover all samples of the above scenarios, we restrict DRM to samples where $\hat{t}_f \neq \hat{t}_s$:

$$
\hat{y} = \begin{cases}
\underset{i}{\arg\max}\, g(h(x; p_{\hat{t}_s}, \theta_{ptm}); \theta_g), & \text{if } \hat{t}_f \neq \hat{t}_s, \\
\underset{i}{\arg\max}\, g(h(x; p_{\hat{t}_f}, \theta_{ptm}); \theta_g), & \text{otherwise.}
\end{cases}
\tag{2}
$$

For samples with $\hat{t}_f \neq \hat{t}_s$, if $\hat{t}_s$ is the correct task identity, it can directly improve TII. If $\hat{t}_s$ is incorrect, the worst result is to predict the wrong class like $\hat{t}_f$. Therefore, this design does not compromise the final performance.

### 3.2.2 Confidence-based Re-matching

However, if $\hat{t}_f$ is identical to $\hat{t}_s$, directly replacing $\hat{t}_f$ with $\hat{t}_s$ for re-matching would be meaningless. Considering that each group of parameters is trained using images associated with its corresponding task identity, the classifier often outputs an unconfident predicted distribution for mismatched samples during inference [26]. Hence, we further propose an *confidence-based re-matching* (CRM) strategy. Initially, we calculate the confidence $E(\tilde{d}(x))$ by function $E(\cdot)$. Given advanced performance and advantages in detecting semantic drift , the post-hoc generalized entropy (GEN) function [51] is applied. GEN calculates generalized entropy [13, 22] only relies on $\tilde{d}(x)$ without any training:

$$E(\tilde{d}(x)) = G_\gamma(\mathbf{d}^*) = -\sum_{c_j} d_{c_j}^{*\,\gamma} \left(1 - d_{c_j}^*\right)^\gamma, \tag{3}$$

where $\gamma$ is hyperparameter, $\mathbf{d}^*$ is the first $M$ values after sorting the probability values in $\tilde{d}(x)$ from descending order, $M$ represents the top-M highest probability, $\tilde{d}(x)$ is obtained by $\hat{t}_s$ in DRM. Subsequently we assess whether a mismatching may occurr for $x$ by the matching detector $De(x)$:

$$De(x; p_{\hat{t}_f}, \theta_{ptm}, \theta_g) = \begin{cases} 0, & \text{if } E(\tilde{d}(x)) \leq \tau, \\ 1, & \text{otherwise,} \end{cases} \tag{4}$$

where $\tau$ is the threshold. When the task identity is inaccurately predicted, $De(x)$ classifies it as 0.

Having detected mismatched samples, *i.e.*, $De(x) = 0$, the critical challenge lies in identifying a more suitable parameter from $P$. Given that it is impractical to establish a definitive boundary between mismatched and correctly matched samples by $\tau$, we refrain from directly replacing $\hat{t}_f$ with another task identity. Rather, we compare the confidence of final predicted distributions yielded by the top-$N$ identities $\Gamma$ in initial matching. Specifically, we first obtain the top-$N$ classes with the highest probabilities from the initial matching prediction distribution $\hat{d}(x)$:

$$\Gamma = \underset{\{c_j\}_{j=1}^N}{\mathrm{argmax}} \; \hat{d}_{c_j}(x), \tag{5}$$

where $\Gamma$ is the set of the top-$N$ classes. Candidate task identities are obtained by converting all classes to task identity in $\Gamma$ with $\mathcal{T}$. Then, we compare the confidence scores $E$ of the final prediction distributions generated by inferencing with the parameters $p$ corresponding to all task identities. The task identity with the highest confidence is selected as the corrected task identity $\hat{t}_s$:

$$\hat{t}_s = \underset{i \in \Gamma}{\arg\max} \; E(g(h(x; p_{\mathcal{T}(i)}, \theta_{ptm}); \theta_g)). \tag{6}$$

The final class prediction $\hat{y}_{CRM}$ is derived from the prediction distribution corresponding to $\hat{t}_s$:

$$\hat{y}_{CRM} = \underset{i}{\mathrm{argmax}} \; g(h(x; p_{\hat{t}_s}, \theta_{ptm}); \theta_g). \tag{7}$$

Finally, the classification outcome $\hat{y}_{CRM}$ associated with the task identity of highest confidence is employed as the re-matching result, thereby diminishing the effects of erroneous detections. Since additional forwards will be introduced when comparing confidence, N is set to be 2. The discussion of N is in the Supplementary Material.

### 3.3 Cross-Task Instance Relationship Distillation

**Cross-task instance relationship distillation** (CTIRD) aims to facilitate the acquisition of task-invariant knowledge across different task parameters. We hope that more samples can be predicted as correct classes or task identities of classes, despite the incorrect initial matching. For the current task $t$, new parameters $p_t$ and old parameters $\{p_1, \ldots, p_{t-1}\}$ of past tasks are available. Combining arbitrary parameter $p_k \in P$ with the pre-trained model $h_{ptm}$, we can acquire the corresponding class token as feature embedding $e_k(x) = h(x; \theta_{ptm}, p_k)$ for $x \in D_t$. To represent new knowledge, we

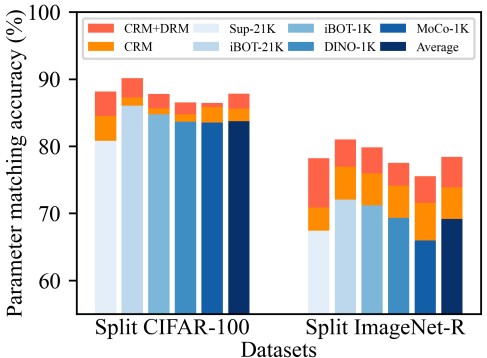
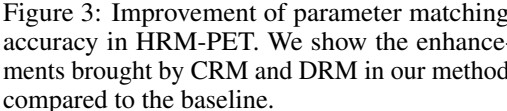
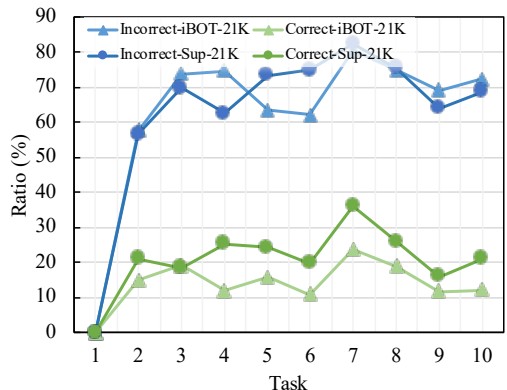

Figure 3: Improvement of parameter matching accuracy in HRM-PET. We show the enhancements brought by CRM and DRM in our method compared to the baseline.

Figure 4: Proportion of detected incorrect (blue) and correct matching (green) samples by confidence score from task1 to task10 for Split ImageNet-R in Sup-21K and iBOT-21K.

extract features for each sample in the current task $t$ by task parameters $p_t$, and then calculate the feature similarity between them to obtain similarity matrix $W_t$:

$$W_t^{ij} = e_t(x_i) \cdot e_t(x_j). \tag{8}$$

During representing old knowledge, images of the current task do not align with any previous tasks, rendering it infeasible to ascertain the most appropriate historical parameters for each current sample. To explore useful knowledge from past parameters, we utilize trained task classifier $g_\omega$ to identify the appropriate, albeit suboptimal, old parameters for each current sample. To incorporate more previous tasks, we select the top-K task identities:

$$\mathbf{s}_i = \operatorname*{argmax}_{\{\mathcal{T}(c_j)\}_{j=1}^K} \hat{d}_{c_j}(x_i),$$
$$S_l = \{s_1^l, s_2^l, s_3^l, \ldots, s_B^l\}, \tag{9}$$

where $\mathbf{s}_i$ represents a subset of top-K task identities selected for $x_i$. For a batch of samples, we can obtain K sets $\{S_1, S_l, \ldots, S_K\}$, each of which represents an identity set with the $l_{th}$ ($l \in [1, K]$) highest confidence level of these samples. For example, $s_2^3$ represents the identity of the second sample in a batch with the third highest confidence in $\mathbf{s}_2$. Then, we can obtain a set $\{W_{S_1}, W_{S_2}, \ldots, W_{S_K}\}$ of similarity matrices recording instance relationships in different feature spaces. To ensure the consistency of shared knowledge across tasks, we compute the Kullback-Leibler (KL) divergence between the normalized instance relationship matrix $W_t$ of the current task and others as a loss:

$$W_{S_l}^{ij} = e_{s_i^l}(x_i) \cdot e_{s_j^l}(x_j),$$
$$\mathcal{L}_{\text{CTIRD}}(p_t, \theta_g) = \sum_{l=1}^K KL(Norm(W_t), Norm(W_{S_l})). \tag{10}$$

Consequently, the final loss function is:

$$\mathcal{L}_{\text{WTP}}(p_t, \theta_g) = \mathcal{L}_{\text{CE}}(p_t, \theta_g) + \lambda_{CT} \mathcal{L}_{\text{CTIRD}}(p_t, \theta_g). \tag{11}$$

The $\lambda_{CT}$ calibrates the trade-off between the acquisition of task-invariant and task-specific expertise.

# 4 Experiments

## 4.1 Datasets

We first carry out experiments on two widely used datasets in CIL performance evaluation: **Split CIFAR-100** and **Split ImageNet-R**. Split CIFAR-100 splits CIFAR-100[41] dataset into 10 tasks. However, CIFAR-100 is relatively easy to classify and is susceptible to information leakage when utilizing some pre-trained models. Split ImageNet-R [88] is a recently proposed, more challenging

Table 1: Results of Split CIFAR-100, Split ImageNet-R, ImageNet-A, and 5-Datasets on five different pre-trained models (PTM). We report the mean and standard deviation of both average final accuracy $A_N$ and average forgetting $F_N$ computed across three random seeds.

| PTM | Method | Split CIFAR-100 | | Split ImageNet-R | | ImageNet-A | | 5-Datasets | |
|---|---|---|---|---|---|---|---|---|---|
| | | $A_N \uparrow$ | $F_N \downarrow$ | $A_N \uparrow$ | $F_N \downarrow$ | $A_N \uparrow$ | $F_N \downarrow$ | $A_N \uparrow$ | $F_N \downarrow$ |
| Sup-21K | DualPrompt [88] | 86.60±0.19 | 4.45±0.16 | 68.79±0.31 | 4.49±0.14 | 39.76±0.18 | 5.85±1.14 | 76.09±0.91 | 15.70±1.13 |
| | S-Prompt++ [86] | 88.81±0.18 | 3.87±0.05 | 69.68±0.12 | 3.29±0.05 | 39.10±0.30 | 5.44±0.98 | 83.38±0.29 | 4.11±0.29 |
| | CODA-Prompt [73] | 86.94±0.63 | 4.04±0.18 | 70.03±0.47 | 5.17±0.22 | 30.96±0.14 | 5.58±0.45 | 62.54±1.34 | 18.58±1.67 |
| | LAE [19] | 85.10±0.24 | 5.01±0.35 | 72.25±0.62 | 3.43±0.65 | 40.36±1.26 | 9.33±0.45 | 64.16±0.10 | 10.25±0.22 |
| | CPrompt [21] | 83.18±1.89 | 6.98±1.56 | 69.24±1.31 | 5.42±1.78 | 37.60±1.01 | 8.12±0.99 | 71.72±0.65 | 18.87±1.12 |
| | InfLoRA [47] | 85.28±0.10 | 4.28±0.91 | 72.24±0.54 | 3.20±1.27 | 42.12±1.97 | 6.01±1.71 | 71.97±0.67 | 12.23±1.05 |
| | HiDe-LoRA [80] | 88.43±0.38 | **3.16±0.16** | 71.90±0.22 | 4.33±0.24 | 42.18±0.10 | 6.15±0.17 | 93.25±0.03 | 1.04±0.03 |
| | HRM-PET (Ours) | **89.45±0.23** | 3.83±0.13 | **73.86±0.14** | 3.60±0.15 | **44.28±0.12** | 5.41±0.32 | **93.99±0.12** | **0.61±0.14** |
| iBOT-21K | DualPrompt [88] | 78.76±0.23 | 9.84±0.24 | 54.55±0.53 | 5.38±0.70 | 21.60±1.70 | 6.92±0.84 | 67.60±1.47 | 26.24±1.11 |
| | S-Prompt++ [86] | 79.14±0.65 | 9.17±1.33 | 55.16±0.83 | 4.07±0.16 | 23.89±0.86 | 5.86±0.68 | 71.46±2.94 | 19.38±4.28 |
| | CODA-Prompt [73] | 80.83±0.27 | 7.50±0.25 | 61.22±0.35 | 9.66±0.20 | 28.10±0.28 | 6.33±0.76 | 56.12±3.18 | 27.23±2.51 |
| | LAE [19] | 78.87±0.58 | 10.55±0.40 | 60.68±0.12 | 4.67±0.87 | 29.76±0.16 | 9.77±1.06 | 63.86±1.00 | 13.75±1.12 |
| | CPrompt [21] | 82.34±0.81 | 6.98±1.16 | 64.64±0.87 | 7.06±0.71 | 31.66±0.61 | 10.26±0.79 | 67.38±1.90 | 16.48±0.81 |
| | InfLoRA [47] | 87.95±0.39 | 4.88±0.46 | 70.13±0.24 | 4.90±0.37 | 33.87±1.55 | 9.04±1.46 | 75.64±0.17 | 8.83±0.59 |
| | HiDe-LoRA [80] | 88.44±0.17 | 4.53±0.34 | 73.40±0.22 | 3.77±0.41 | 38.55±0.43 | 7.49±0.19 | 93.53±0.31 | 1.69±0.10 |
| | HRM-PET (Ours) | **89.70±0.13** | **3.75±0.11** | **75.23±0.21** | 3.68±0.17 | **40.88±0.32** | 4.98±0.61 | **94.38±0.11** | 1.07±0.15 |
| iBOT-1K | DualPrompt [88] | 76.63±0.05 | 8.41±0.40 | 61.51±1.05 | 5.02±0.52 | 21.60±0.39 | 6.43±0.64 | 65.35±1.26 | 27.62±1.75 |
| | S-Prompt++ [86] | 77.53±0.56 | 8.07±0.97 | 60.82±0.68 | 4.16±0.14 | 21.12±1.12 | 6.52±0.37 | 71.71±1.50 | 6.64±0.14 |
| | CODA-Prompt [73] | 79.11±1.02 | 7.69±1.57 | 66.56±0.68 | 7.22±0.38 | 28.69±0.73 | 6.34±1.23 | 44.34±3.33 | 21.89±4.06 |
| | LAE [19] | 75.45±0.43 | 10.55±0.36 | 67.95±0.55 | 5.56±0.41 | 27.18±0.21 | 9.77±0.67 | 67.00±1.46 | 13.75±0.98 |
| | CPrompt [21] | 76.32±0.79 | 12.49±1.36 | 68.25±1.43 | 8.25±1.19 | 29.70±0.91 | 8.23±1.11 | 75.21±0.84 | 9.86±1.12 |
| | InfLoRA [47] | 85.51±0.10 | 6.28±0.16 | 71.90±0.10 | 4.91±2.70 | 32.01±0.70 | 7.45±0.67 | 75.33±0.61 | 9.12±0.48 |
| | HiDe-LoRA [80] | 86.43±0.23 | 5.23±0.34 | 72.79±0.28 | 4.39±0.16 | 35.63±0.25 | 6.25±0.13 | 93.48±0.06 | 1.70±0.03 |
| | HRM-PET (Ours) | **87.11±0.15** | **4.57±0.14** | **74.64±0.28** | 3.92±0.19 | **37.89±0.34** | 5.19±0.41 | **94.15±0.15** | 0.87±0.20 |
| DINO-1K | DualPrompt [88] | 74.90±0.21 | 10.26±0.62 | 58.57±0.45 | 5.80±0.21 | 21.27±0.52 | 5.16±0.81 | 68.21±1.11 | 24.04±0.86 |
| | S-Prompt++ [86] | 74.97±0.46 | 7.78±0.66 | 57.64±0.16 | 5.08±0.31 | 20.58±1.52 | 6.22±0.68 | 75.19±0.80 | 6.76±1.76 |
| | CODA-Prompt [73] | 77.50±0.64 | 8.10±0.01 | 63.15±0.39 | 6.86±0.11 | 24.28±0.56 | 6.66±1.22 | 51.52±2.10 | 27.55±3.01 |
| | LAE [19] | 73.62±0.44 | 12.56±0.29 | 63.70±0.84 | 5.44±0.24 | 23.10±0.38 | 8.68±1.01 | 65.33±1.55 | 10.25±1.23 |
| | CPrompt [21] | 74.64±1.69 | 11.79±1.99 | 63.67±0.57 | 9.85±0.91 | 26.49±0.81 | 7.80±0.87 | 73.40±0.95 | 17.67±1.41 |
| | InfLoRA [47] | 82.37±0.65 | 6.40±0.33 | 68.51±0.20 | 5.01±0.69 | 31.67±0.28 | 5.12±0.49 | 79.09±0.23 | 11.23±0.05 |
| | HiDe-LoRA [80] | 84.85±0.21 | 5.36±0.22 | 70.42±0.22 | 4.60±0.19 | 31.72±0.33 | 4.98±0.34 | 93.25±0.04 | 1.34±0.01 |
| | HRM-PET (Ours) | **85.93±0.14** | 5.14±0.24 | **72.32±0.20** | 3.49±0.27 | **33.37±0.35** | 4.78±0.55 | **93.90±0.05** | **0.67±0.07** |
| MoCo-1K | DualPrompt [88] | 77.77±0.68 | 6.61±1.08 | 52.57±0.82 | **2.73±0.49** | 18.07±1.03 | **4.11±0.65** | 68.17±0.35 | 23.56±0.46 |
| | S-Prompt++ [86] | 76.30±0.54 | 14.67±0.64 | 53.15±1.10 | 4.11±1.84 | 17.96±0.66 | 4.90±1.96 | 71.75±0.57 | 13.60±0.73 |
| | CODA-Prompt [73] | 76.83±0.34 | 12.60±0.02 | 55.75±0.26 | 10.46±0.04 | 13.75±0.21 | 8.34±0.19 | 52.81±2.41 | 33.58±2.99 |
| | LAE [19] | 78.31±0.32 | 15.24±0.19 | 54.11±0.16 | 10.21±0.34 | 21.42±0.15 | 12.43±1.11 | 57.44±0.18 | 17.11±0.36 |
| | CPrompt [21] | 77.38±0.99 | 12.24±1.56 | 59.47±2.56 | 8.61±1.85 | 22.16±0.91 | 6.94±0.69 | 73.17±0.84 | 17.67±1.34 |
| | InfLoRA [47] | 82.70±0.39 | 8.38±0.66 | 65.03±0.87 | 5.69±1.10 | 26.00±0.81 | 6.50±0.64 | 74.00±0.38 | 10.49±0.42 |
| | HiDe-LoRA [80] | 85.37±0.21 | 5.42±0.19 | 68.01±0.59 | 4.57±0.86 | 28.30±0.47 | 6.74±0.35 | 92.72±0.07 | 1.71±0.06 |
| | HRM-PET (Ours) | **86.00±0.14** | 5.32±0.19 | **69.32±0.22** | 4.10±0.35 | **29.73±0.52** | 6.23±0.61 | **93.31±0.08** | 1.23±0.15 |

Table 2: Comparisons with other methods relying on pre-trained models. To ensure a fair comparison, we re-implemented our method under the same weights in SLCA [100].

| Method | Split CIFAR-100 | | Split ImageNet-R | | ImageNet-A | | 5-Datasets | |
|---|---|---|---|---|---|---|---|---|
| | $A_N \uparrow$ | $F_N \downarrow$ | $A_N \uparrow$ | $F_N \downarrow$ | $A_N \uparrow$ | $F_N \downarrow$ | $A_N \uparrow$ | $F_N \downarrow$ |
| ESN [87] | 86.34±0.52 | 4.76±0.14 | 65.83±0.58 | 5.93±0.31 | 45.08±0.23 | **6.25±0.84** | 85.71±1.47 | 2.85±0.61 |
| SLCA [100] | 91.53±0.28 | 5.33±0.31 | 77.00±0.33 | 4.34±0.32 | 52.85±0.40 | 7.72±0.65 | 85.24±0.84 | 11.32±0.92 |
| HRM-PET(ours) | **91.55±0.63** | **3.83±0.13** | **79.03±0.14** | **3.11±0.15** | **55.43±0.12** | 7.89±0.32 | **94.12±0.12** | **0.58±0.14** |

dataset. It collects 30000 images of different styles and is difficult to differentiate for general classification models from ImageNet [14], which is similarly split into 10 tasks.

To investigate performance on datasets with the larger domain gap from the pre-trained ImageNet, we evaluate HRM-PET on **ImageNet-A** [27], consisting of images that are misclassified by ResNet models. It is also divided into 10 tasks and 20 classes for every task [59]. Moreover, **5-Datasets** [15] consisting of CIFAR-10 [41], MNIST [42], Fashion-MNIST [91], SVHN [62] and notMNIST [4] is examined, which are divided into 5 tasks according to different datasets.

## 4.2 Evaluation Settings

**Comparing methods.** We firstly compare multiple PET-based methods: DualPrompt [88], S-Prompt++ [86], CODA-Prompt [73], LAE [19], CPrompt [21], InfLoRA [47], and HiDe-LoRA [80, 81]. CODA-Prompt, LAE, and InfLoRA do not maintain task-specific parameters and do not require task identification. For fairness, both our method and HiDe-Prompt use LoRA as the PET, and therefore HiDe-Prompt is referred to as HiDe-LoRA in the table. The LoRA modules are integrated into the initial five layers of the ViT. Moreover, we compare the performance of different pre-trained

Table 3: Ablation experiment to prob DRM, CRM, and CTIRD in ImageNet-R.

| Method | Sup-21K | iBOT-21K | iBOT-1K | DINO-1K | MoCo-1K |
|---|---|---|---|---|---|
| Baseline | 71.47±0.26 | 73.49±0.31 | 72.99±0.56 | 70.40±0.62 | 67.89±0.29 |
| Baseline+DRM | 72.55±0.10 | 74.11±0.22 | 73.48±0.35 | 71.05±0.27 | 68.70±0.42 |
| Baseline+CRM | 72.80±0.15 | 74.35±0.26 | 74.01±0.30 | 71.32±0.23 | 68.80±0.20 |
| Baseline+DRM+CRM | 73.38±0.21 | 74.80±0.15 | 74.18±0.18 | 71.85±0.45 | 69.10±0.15 |
| Baseline+DRM+CRM+CTIRD | 73.86±0.14 | 75.23±0.21 | 74.64±0.28 | 72.32±0.20 | 69.32±0.22 |

Table 4: The additional computational time during the inference process. We present the Time (ms) per image on the test set, and $A_N(\%)$, after the application of DRM and CRM.

| Method | CIFAR-100 | | ImageNet-R | |
|---|---|---|---|---|
| | $A_N \uparrow$ | Time↓ | $A_N \uparrow$ | Time↓ |
| Baseline | 88.40 | 2.69 | 71.60 | 2.81 |
| Baseline+DRM | 88.80 | 2.81 | 72.60 | 2.91 |
| Baseline+CRM+DRM | 89.45 | 3.01 | 73.86 | 3.24 |

Table 5: Comparison of different knowledge distillation methods under Sup-21K and iBOT-21K pre-trained model.

| Distillation | CIFAR-100 | | ImageNet-R | |
|---|---|---|---|---|
| | Sup-21. | iBOT-21. | Sup-21. | iBOT-21. |
| Logits [45] | 87.66 | 88.02 | 72.88 | 74.51 |
| Features [105] | 87.40 | 83.72 | 71.15 | 71.64 |
| IRD* [7] | 88.59 | 88.98 | 73.21 | 74.58 |
| CTIRD | **89.45** | **89.70** | **73.86** | **75.23** |

models: Sup-21K [68], iBOT-1K [104], iBOT-21K [104], DINO-1K [6] and MoCo-1K [25]. We employ the same supervised or self-supervised pre-trained ViT-B/16 as the backbone. In Sup-21K, for fair comparison, we report the results under **uniform checkpoint**[1]. More details are shown in Supplementary Material. Additionally, we conduct comparisons with SLCA [100] and ESN [87]. Follow [73], the performance is measured by average final accuracy $A_N$ and average forgetting $F_N$ after learning $N$ tasks [53, 9, 88].

## 4.3 Comparison with the State-of-the-art Methods

To evaluate the effectiveness, we perform extensive comparative experiments with 7 state-of-the-art PET-based methods. This comparison involves five pre-trained models across four datasets: Split-CIFAR100, Split ImageNet-R, ImageNet-A, and 5-Datasets. Subsequently, we present comparative results with 2 sota methods based on the same pre-trained models but not designed with PET.

The results are shown in Table 1. Overall, our method achieves state-of-the-art results under 20 settings of four datasets with various pre-trained backbones. On the Split ImageNet-R and ImageNet-A with large intra-class diversity [103, 88], HRM-PET shows substantial performance improvements. It is worth noting that our method improves 1.61% and 2.10% with the model pre-trained on Sup-21K, 1.83% and 2.33% with the model pre-trained on iBOT-21K. Moreover, HRM-PET also brings improvement on Split CIFAR-100. On 5-Datasets, characterized by large task gaps, our method achieves improvements of 0.74%, 0.85%, 0.67%, 0.65%, 0.59% respectively under different backbones. This indicates that HRM-PET consistently achieves better performance, and the improvement is particularly significant on challenging datasets.

In addition to applying PET in CIL, there exist many other methods, such as direct fine-tuning. We present the comparison between HRM-PET and two representative algorithms in Table 2. On the one hand, considering $A_N$, HRM-PET surpasses other methods 2.03% and 2.58% on Split ImageNet-R and ImageNet-A. Moreover, our method achieves 8.41% improvements on 5-Datasets. The results demonstrate the superiority of HRM-PET. On the other hand, although our method only tunes a few parameters, HRM-PET stably exhibits competitive performance on all datasets.

## 4.4 Ablation Study

**Effectiveness of the proposed components.** As illustrated in Table 3, we conduct a comprehensive analysis of each module, across five pre-trained models on the Split ImageNet-R datasets. $A_N$ is utilized as measurement. The baseline is trained with in Section 3.1. According to the results, we have the following observations: 1) Compared with baseline model, both CRM and DRM calibrate incorrect parameter matching, hence improving performance. Furthermore, CRM aims to detect all mismatched samples based on confidence, resulting in powerful performance. 2) Combining CRM and DRM achieves better performance compared with either CRM or DRM. As shown in Fig. 1b, there is a partial intersection between correct and incorrect matching in the confidence-based distribution. DRM compensates for the limitations on these samples, thereby combining CRM

---

[1]ViT-B_16.npz

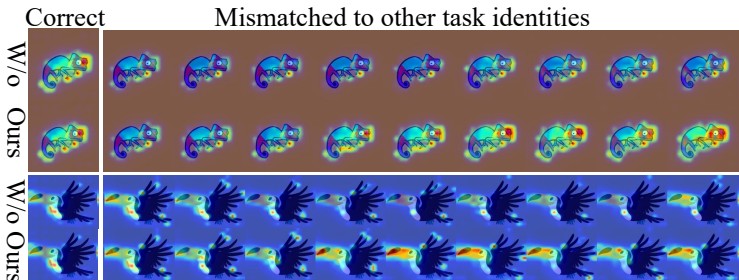

Figure 5: Visualization of attention regions with (Ours) and without CTIRD (W/o). "Correct" means the parameters with the correct task identity. "Mismatched to other task identities" means parameters corresponding to other incorrect task identities.

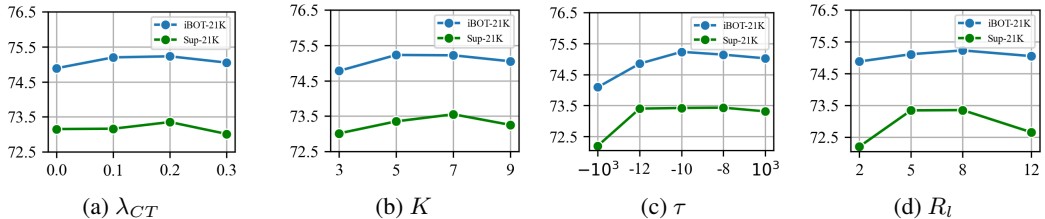

(a) $\lambda_{CT}$        (b) $K$        (c) $\tau$        (d) $R_l$

Figure 6: Variation of $A_N$ with hyperparameters. We show influences of 4 hyperparameters within the context of the ImageNet-R dataset with the Sup-21K and iBOT-21K setting: (a) $\lambda_{CT}$, (b) $K$, (c) $\tau$, (d) $R_l$.

and DRM further improves the matching and classification performances. 3) By combining all components, the model attains the best $A_N$, which demonstrates the complementarity of the proposed re-matching and distillation strategies.

**Parameter matching accuracy.** To prob the effect of re-matching strategies on matching accuracy, we conduct extensive experiments in Fig. 3. First, on the Split ImageNet-R with large intra-class diversity [88], our re-matching improves matching accuracy by an average of 9.28% using various pre-trained backbones. Particularly, we achieve at least 2.91% in various settings, which demonstrates the robustness of the re-matching. Besides, on the Split CIFAR-100 with matching accuracy exceeding 80%, combining DRM and CRM still brings an average matching accuracy improvement of 4.13%. After obtaining higher matching accuracy, HRM-PET achieves higher $A_N$ in Table 1.

**Confidence-based detection.** To validate the CRM's ability to detect mismatching, we show the proportion of incorrectly and correctly matched samples filtered by the confidence threshold. In all experiments, the threshold $\tau$ is consistently set as -10. As depicted in Fig. 4, with the help of the confidence-based threshold, CRM is able to detect the most mismatched samples during the incremental process. Note that about 10-30% of correctly matched images are inevitably incorporated. To handle these data, we design equation 6 to determine whether to replace the task identity.

**Inference time.** When re-matching is conducted based on the predicted distribution, the model needs an additional forward process ($N = 2$), which leads to an increase in the inference time. Table 4 shows the increased consumption. Results are fairly compared on RTX3090 at the same batch size. Particularly, DRM and CRM only deal with a part of samples that are likely to be mismatched, hence we can observe that the average time cost on the test set is only about 0.4ms. Importantly, significant performance improvements can be achieved with slight time cost. Therefore, our calibrating strategy based on re-matching mechanism is promising for future research.

**Training time.** The training time of HRM-PET is acceptable. In CTIRD, we first utilize the pre-trained task classifier $g_\omega$ to obtain the top-$K$ old task identities with the highest confidence scores for all samples. The time cost of performing a forward through the classifier is negligible. Subsequently, before training, each sample undergoes $K$ inferences to obtain $K$ features for knowledge distillation. Compared to training, the inference-only forward introduces minimal additional time overhead. Experimentally, we measured the training time of baseline and HRM-PET on ImageNet-R on an RTX 3090. Our method incurs only an additional 4.4% (i.e., 0.30 hours) of time overhead.

**Knowledge distillation.** Knowledge distillation [45] is a classic mechanism for preserving model stability. Through experimental exploration, we assess the influence of various knowledge distillation techniques. The compared methods contain traditional logits [45], feature distillation [105], and similar instance relation distillation (IRD) [7] in Table 5. We implement IRD for each old task (IRD*) without our top-K strategy. According to the results, CTIRD surpasses all alternative methods. The previous methods add overly rigid constraints to the model, which impede the plasticity of new task learning. Differently, CTIRD selects appropriate old parameters for each sample by the top-K mechanism, which further prompts the model to learn shared information across tasks.

**Visualization.** In CTIRD, we visualized the attention area with parameters of different task identities in Fig. 5. The first column displays attention maps generated using the correct task identity, while columns 2–10 correspond to attention maps produced using incorrect task identities. Two main observations can be drawn: 1) On the one hand, there is almost no difference in the attention obtained with the correct parameters when CTIRD is used and not used. This suggests that CTIRD does not impair task-specific knowledge. 2) On the other hand, when CTIRD is applied, even if mismatched parameters are attached, the attention still focuses on key feature areas of the object. These key feature areas help them still have the correct task identities or classes in the predicted distribution when the parameters are mismatched. Hence, CTIRD improves shared knowledge.

### 4.5 Hyperparameter Analysis

In this section, the effect of hyperparameters on the HRM-PET is investigated. All experiments are conducted on Split ImageNet-R, for Sup-21k and iBOT-21k pertaining settings. First, we prob the effect of the CTIRD loss coefficient $\lambda_{CT}$ on $A_N$, which is shown in Fig. 6a. The performance of HRM-PET improves as $\lambda_{CT}$ increases and achieves the best performance when $\lambda_{CT} = 0.2$. A smaller $\lambda_{CT}$ compromises the PET's capacity to acquire task-invariant knowledge. Conversely, an excessively large $\lambda_{CT}$ would obstruct the acquisition of task-specific knowledge in new tasks. Next, We also present the influence of the number $K$ for CTIRD in Fig. 6b. $K$ controls the amount of knowledge distillation with past task parameters, which influences the trade-off between task-specific and task-invariant knowledge. Note that we utilize the common setting $K = 5$ to reduce the training computational consumption.

In addition, as shown in Fig. 6c, the influence of the confidence threshold $\tau$ in re-matching is explored. A lower $\tau$ calibrates fewer mismatched samples, whereas a higher threshold may classify correctly matched samples as mismatched, we set $\tau$ as -10. As validated in Fig. 3, combining confidence-based re-matching strategy, the accuracy of task identity matching can be significantly improved. Finally, we explore the effect hyperparameter $Rank$ $(R_l)$ leveraged in LoRA, which is shown in Fig. 6d. In HRM-PET, $R_l = 8$ is suitable to learn task-specific knowledge. For more details, please see the Supplementary Material.

## 5 Conclusion

In this paper, we propose HRM-PET for rehearsal-free continual learning with parameter-efficient tuning. We design a hybrid re-matching mechanism based on the initial predicted distribution to improve parameter matching accuracy in PET-based CL methods. Specifically, the direct re-matching handles some samples by replacing the incorrect initial task identity with the correct one from prediction. In addition, the confidence-based re-matching addresses more mismatched samples without correct task identity in the predicted distribution. Furthermore, we introduce cross-task instance relationship distillation to reduce the dependence of the model on matching accuracy. Minimizing hyperparameter dependency and improving inference efficiency represent important avenues for future research. Overall, extensive comparison and ablation experiments indicate that HRM-PET achieves state-of-the-art performance.

## 6 Acknowledgments

This work was supported by the Natural Science Foundation of Tianjin, China (No.24JCZXJC00040), Shenzhen Science and Technology Program (No. JCYJ20240813114229039), the National Natural Science Foundation of China (No. 623B2056, 624B2072), the Fundamental Research Funds for the Central Universities, the Supercomputing Center of Nankai University (NKSC).

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
