# OpenReview forum: "Hybrid Re-matching for Continual Learning with Parameter-Efficient Tuning"
_NeurIPS.cc/2025/Conference — NeurIPS 2025 poster_

### Official Review · Reviewer_4szz · 2025-06-13

**Clarity:** 2
**Significance:** 3
**Originality:** 3
**Rating:** 4
**Confidence:** 4

**Summary:**

This work proposes a hybrid re-matching method called HRM-PET for rehearsal-free continual learning with parameter-efficient tuning. Two types of re-matching are integrated into a whole framework. The direct re-matching is straightforward by replacing the incorrect initial task identity with the different predicted task identity. The confidence-based re-matching handles the more challenging case that direct re-matching fails to improve. The mismatched samples with wrong task identity are detected with a confidence threshold, and then the task identity of highest confidence is chosen from top-N identities. Moreover, cross-task instance relationship distillation is used to acquire task-invariant knowledge. Experiments on four datasets under five pre-trained settings demonstrate the competitive performance of HRM-PET.

**Questions:**

1. Fig. 5. I can not understand the underlying message what Fig. 5 wants to convey. Which one is better, and why?
2. Fig. 4. I don't know what this figure aims to showcase.
3. Fig. 2 appears to be a little mess. And where is the module of instance relationship distillation?
4. Although re-matching improves the experiment performance for continual learning, is there any explaination or theory? Why can this method work?
5. Can the re-matching procedure be casted in more reasonable framework, such as probabilistic reasoning, or VAE (I don't know if it can work, just mention some) something like that? Currently this method appears to be like "if-then-else" rules, lacking deep thoughts.

**Ethical Concerns:**

["NO or VERY MINOR ethics concerns only"]

**Final Justification:**

I thank the authors for answering my concerns and questions. More experimental results were exposed in Author rebuttal, which make me satisfactory to the quality of this work. It appears that the authors' responses solve most of my concerns of this paper, and I would like to maintain my initial rating.

**Limitations:**

See above review for suggestions for improvement.

**Paper Formatting Concerns:**

Not applicable.

**Quality:**

3

**Strengths And Weaknesses:**

Strengths:
1. The experiment evaluation is extensive and solid. Results of tables and figures demonstrate the performance of the proposed method over SOTA algorithms.
2. The re-matching idea between the current example (with its class label prediction) and the task ID appears to be novel. Two kinds of re-matching schemes seem to be complementary to each other. Moreover, cross-task instance relationship distillation enhances the overall performance.

Weaknesses:
1. Clarity. I feel very confused at several places when I read this paper first time. For example, if A and B are matched or re-matched, what are A and B in this hybrid re-matching method? It took me much time to answer such questions. Also, some notations should be clearly defined before using these notations.
2. Quality. Personally I think the techniques used in this work are straightforward and lack some theoretical depth. For example, if \hat{t}_f is not equal to \hat{t}_s, the class label is predicted simply with \hat{t}_s. In the confidence-based rematching, a simple threshold method is adopted. In the instance relationship distillation, the standard meaure of KL divergence is employed. From these aspects, these techniques appears to be not novel.
3. Originality. As above stated, the techniques used in this paper seem to be less original, except the overal re-matching framework.

Minor points:
1. Page 7, line 216. "between HRM-PET and three representive algorithms in Table 2". "three" should be corrected as "two".

---

> ### Author Rebuttal · Authors · 2025-07-30
>
> Thank you for your insightful comments and questions.
>
> **Q1: Clarity**
>
>
> **A1:** For PET-based CL methods, the model first predicts task identity for each sample $x$. Then, the corresponding PET parameters $p_\{\hat{t}_{f}}$ are retrieved from the parameter pool $P$, and are used for subsequent class prediction. When the predicted task identity equals the true task identity, we refer as **matching**. When they differ, it is referred to as **mismatching**.  In our method, we correct the wrong task identity, which we term **re-matching**. We will add the this clarity in the revision.
>
> ---
>
> **Q2&Q3: Quality and Originality**
>
>
> **A2&A3:** Our HRM-PET is simple, yet effective and stable. The main contributions are notable:
>
> Most importantly, we explore the core and long-existing problem in PET-based CL, the potential inaccuracy in the task-parameter matching process during the testing phase.
>
> We propose a hybrid re-matching framework to improve the accuracy of task identity prediction. CTIRD alleviates the reliance on matching, facilitating shared knowledge without compromising task-specific knowledge.
>
> Moreover, empirical results serve as evidence for the rationality and effectiveness of our method.
>
> ---
>
> **Q4: Explanation of Fig. 5**
>
>
> **A4:** Fig. 5 compares the attention regions of the model with (Ours) and without CTIRD (W/o). The first column displays attention maps generated using the correct task identity, while columns 2–10 correspond to attention maps produced using incorrect task identities. Two main observations can be drawn:
>
> 1) For the first column, with CTIRD, the model still focuses on the key regions, indicating that CTIRD does not compromise the plasticity.
>
> 2) For columns 2–10, even when incorrect task parameters are used, the model with CTIRD still attends to the critical features, e.g., the lizard’s head and limbs, the bird’s beak. Hence. CTIRD enhances knowledge sharing and reduces the model’s reliance on task matching, demonstrating the superiority of CTIRD.
>
> ---
>
> **Q5: Explanation of Fig. 4**
>
>
> **A5:** Fig. 4 illustrates the ratio of incorrectly and correctly matched samples that satisfy $\tau$, relative to the total number of actual mismatched and matched samples, respectively, across each task in the incremental sequence. Specifically, the blue curve shows that over 70\% of all mismatched samples are detected by $\tau$, demonstrating that CRM can identify most mismatched samples during the incremental process. Additionally, the green curve indicates that 10–30\% of correctly matched samples are misidentified as mismatches, highlighting the necessity of using Eq. 5 to determine whether task identity replacement is required.
>
> ---
>
>
> **Q6: Instance relationship distillation in Fig. 2**
>
>
> **A6:** The module of instance relationship distillation is in the section with **training** at the bottom of Fig. 2. The loss of instance relationship distillation is between the old feature space and the new feature space. We will add the module name in the revision to improve clarity.
>
> ---
>
> **Q7: Theory of re-matching improving the experiment performance for continual learning**
>
> **A7:** The accuracy of task identity prediction has been shown to be critical for improving CL performance, as demonstrated in HiDe-Prompt [77]. Assuming that in the class incremental learning (CIL) scenario, a total of $T$ tasks are defined as $D = \{D_1, D_2, \ldots, D_T\}$. In $D_t$ of task $t$, $\mathcal{X}\_{t}$ and $\mathcal{Y}\_t$ are the domain and label of task $t$. Let $\mathcal{X}\_t=\bigcup\_{j}\mathcal{X}_{t,j}$ and $ \mathcal{Y}\_t=\{\mathcal{Y}\_{t,j}\}$, where $j \in \{1,...,|\mathcal{Y}\_t|\}$ indicates the $j$-th class in task $t$.
> Given a pre-trained model $f\_\theta$, CIL aims to learn $P(\boldsymbol{x} \in \mathcal{X}\_{i,j}|D,\theta)$ for the sample $\boldsymbol{x} \in \bigcup\_{k=1}^{t}\mathcal{X}\_{k}$, where $j \in \{1,...,|\mathcal{Y}\_t|\}$ indicates the $j$-th class in task $t$. Based on the theorem of Bayes, the goal can be decomposed as:
>
>    $P(\boldsymbol{x} \in \mathcal{X}\_{i,j}|D,\theta) = P(\boldsymbol{x} \in \mathcal{X}\_{i,j}|\boldsymbol{x} \in \mathcal{X}\_{i},D,\theta) P(\boldsymbol{x} \in \mathcal{X}\_{i}|D,\theta) $
>
> where $P(\boldsymbol{x} \in \mathcal{X}\_{i}|D,\theta)$ is the task identity inference probability, $P(\boldsymbol{x} \in \mathcal{X}\_{i,j}|\boldsymbol{x} \in \mathcal{X}\_{i},\mathcal{D},\theta)$ is the within-task prediction probability. Hence, when the performance of task identity inference is improved, the performance of CIL will be enhanced. For more proof details, we recommend referring to HiDe-Prompt.
>
> ---
>
> **Q8: More reasonable framework**
>
> **A8:** We sincerely thank for the insightful suggestion. The idea of framing re-matching within a probabilistic framework is indeed inspiring. Assume the input is $\boldsymbol{x}$. The posterior probability that it belongs to class $y = c$ is $P(y=c | \boldsymbol{x}, D, \theta)$. We can further decompose it into task identity inference and within-task prediction:
>
> $
> \begin{aligned}
> P(y = (i,j) \mid \boldsymbol{x}, D, \theta) = P(y = (i,j) \mid \boldsymbol{x}, \text{task}=i, D, \theta) \cdot P(\text{task}=i \mid \boldsymbol{x}, D, \theta)
> \end{aligned}
> $
>
> where $y = (i,j)$ is the $j$-th class in task $t$. After training with PET, we obtain a parameter pool $\boldsymbol{\mathcal{P}} = \{p_1, p_2, \dots, p_T\}$, where $p_i$ denotes the learned parameters specific to task $i$. Rematching involves inference with multiple $p$. To aggregate these predictions in a principled manner, we draw upon Bayesian Model Averaging (BMA), which supports combining models under uncertainty about model selection. Applying BMA, the final posterior is expressed as:
>
> $
> P(y=(i,j) | \boldsymbol{x}, D, \theta, \boldsymbol{\mathcal{P}}) = \sum_{i=1}^T P(y=j | \boldsymbol{x}, \text{task}=i, \theta, p_i) \cdot P(\text{task}=i | \boldsymbol{x}, D, \theta)
> $
>
> However, not all task parameters can be weighted equally. We hope that the correct task identity has a higher weight. Therefore, we calculate the weights $\phi_t$ by the conditions in our strategy, such as the confidence of CRM:
>
> $
> P(y=(i,j) | \boldsymbol{x}, D, \theta, \boldsymbol{\mathcal{P}}) = \sum_{i=1}^T P(y=j | \boldsymbol{x}, \text{task}=i, \theta, p_i) \cdot P(\text{task}=i | \boldsymbol{x}, D, \theta) \cdot
> \phi_t
> $
>
> In our paper, $\phi_i$ acts as a binary gate (i.e., $\phi_i \in \{0,1\}$). Our rematching is uniformly implemented in the above equation. We will include this insight in the revisions.
>
> ---

---

### Official Review · Reviewer_b4of · 2025-06-20

**Clarity:** 1
**Significance:** 2
**Originality:** 2
**Rating:** 4
**Confidence:** 4

**Summary:**

This paper introduces HRM-PET, a novel replay-free continual learning method using Parameter-Efficient Tuning. To enhance inference-time parameter matching, HRM-PET proposes a Hybrid Re-matching mechanism that intelligently utilizes the model’s prediction distribution. It further integrates Cross-Task Instance Relation Distillation to cultivate task-agnostic knowledge and align feature representations. Comprehensive experiments demonstrate HRM-PET’s superior, state-of-the-art performance with minimal additional inference overhead.

**Questions:**

See weaknesses.

**Ethical Concerns:**

["NO or VERY MINOR ethics concerns only"]

**Final Justification:**

I thank the authors for their detailed rebuttal, which has effectively addressed my primary concerns. While I believe the paper could be further strengthened with more robust theoretical validation and additional refinement of the writing, the clarifications provided are sufficient. Accordingly, I have raised my score to 4.

**Limitations:**

The proposed method may incur substantial computational overhead.

**Paper Formatting Concerns:**

No major formatting issues were observed in the paper.

**Quality:**

2

**Strengths And Weaknesses:**

### **Strengths**

1. The paper proposes an innovative and effective Hybrid Re-matching strategy (DRM & CRM) that cleverly leverages the prediction distribution to address the critical inference-time parameter matching inconsistency in PET methods.
2. The paper provides comprehensive experimental validation and extensive ablations.
3. The code is available

### **Weaknesses**

1. The paper lacks precise explanations for subscript notations in equations (3), (5), and (7), leading to comprehension difficulties. Specifically, $\tilde{d}(x)$ appears to denote a class-level distribution, while $\hat{d}(x)$ seems to represent a task-level distribution. However, the subscript representations in equation (5) and (7) vary, requiring explicit clarification for the meaning of each subscript.
2. It is unclear which prompt/parameter (e.g., $\hat{t}_f$ or $\hat{t}_s$) is utilized to obtain $\tilde{d}(x)$ in equation (3). A clear statement specifying the parameter source for this prediction distribution is essential for reproducibility and understanding.
3. While inference time is discussed, the paper lacks a comprehensive analysis of training time, which appears to significantly increase due to multiple additional forward passes.
4. The ablation study does not sufficiently clarify the independent contributions and necessity of each re-matching module.
5. Given that CRM appears to be an independent module from DRM, the rationale for performing DRM prior to CRM, rather than directly applying CRM for all mismatches, remains unclear and warrants further explanation or empirical validation.
6. The paper exhibits several technical and typographical errors that detract from its formal presentation and readability. For instance, each equation should conclude with appropriate punctuation (e.g., a period or comma). Additionally, on line 115, "(LoRA)" appears to be incorrect. The phrase "Forfair, LoRA is employed as PET in HiDe-Prompt i.e. HiDe-LoRA and our method." on line 193 is unclear and grammatically awkward, requiring rephrasing for clarity.

---

> ### Author Rebuttal · Authors · 2025-07-30
>
> Thank you for your insightful comments and questions.
>
> **Q1: Explanations for subscript notations and distribution**
>
> **A1:** Clarification of subscript: $\hat{d}(x)$ and $\tilde{d}(x)$ are all class-level distribution. $\mathcal{T}$ is used to convert class in $\hat{d}(x)$ into task identity in equations 5 and 7.
>
> Clarification of distribution: $j$ in equation 3, $c_j$ in equations 5 and 7 all represent the index of the category.  We will unify the expression in revision.
>
> ---
>
> **Q2: A clear statement specifying the parameter source for this prediction distribution**
>
> **A2:** In fact, we utilize $\hat{t}\_{s}$ in DRM to obtain $\tilde{d}(x)$ in equation 3. DRM has corrected part of the task identities, using $\hat{t}_{s}$ can reduce subsequent repeated calculations.
>
> ---
>
> **Q3: A comprehensive analysis of training time**
>
> **A3:** During training, for each new task $t$, we compute the feature embedding $ e_k(x) $ for each sample under task $t$ using the top-$K$ ($K=5$) previous task-specific parameters. Specifically, we first utilize the pre-trained task classifier $g_{\omega}$ to obtain the top-$K$ old task identities with the highest confidence scores for all samples. The time cost of performing a forward through the classifier is negligible.
>
> Subsequently, before training, each sample undergoes $K$ inferences to obtain $K$ features for knowledge distillation. Compared to training, the inference-only forward introduces minimal additional time overhead.
>
> Experimentally, we measured the training time of baseline and HRM-PET on ImageNet-R on an RTX 3090 with a batch size of 64.  As shown in the table, our method incurs only an additional 4.4\% (i.e., 0.30 hours) of time overhead, which is acceptable. We will add the discussion in the revision.
>
> | Method        | Baseline    | HRM-PET (ours) |
> |---------------|-------------|----------------|
> | Training Time | 6.75 hours  | 7.05 hours     |
>
> ---
>
> **Q4: The independent contributions and necessity of each re-matching module in the ablation study.**
>
> **A4:** As shown in the table, we conduct more detailed ablations on ImageNet-R. We have the following observations:
>
> 1) Compared with the baseline, each re-matching module brings improvements, which demonstrates independence.
>
> 2) Combining CRM and DRM achieves better performance compared with either CRM or DRM, which shows the necessity of each re-matching module.
>
> | Method                     | Sup-21K | iBOT-21K | iBOT-1K | DINO-1K | MoCo-1K |
> |----------------------------|---------|----------|---------|---------|---------|
> | Baseline                   | 71.52   | 73.60    | 72.98   | 70.43   | 67.98   |
> | Baseline+DRM               | 72.60   | 74.09    | 73.58   | 71.12   | 68.69   |
> | Baseline+CRM               | 72.80   | 74.35    | 74.01   | 71.32   | 68.80   |
> | Baseline+CTIRD             | 72.18   | 74.08    | 73.64   | 71.04   | 68.12   |
> | Baseline+DRM+CRM           | 73.40   | 74.75    | 74.18   | 71.85   | 69.10   |
> | Baseline+DRM+CRM+CTIRD     | 73.86   | 75.23    | 74.64   | 72.32   | 69.32   |
>
> ---
>
> **Q5: The rationale for performing DRM prior to CRM**
>
> **A5:** As shown in the table, we conduct ablations on the relationship between CRM and DRM. We make the following validation:
>
> 1) Combining CRM and DRM achieves better performance than using either DRM or CRM alone, demonstrating their complementarity.  In Figure 1 b, there is a partial intersection between correct and incorrect matching in the confidence-based distribution.  DRM compensates for the limitations on these samples, thereby combining CRM and DRM further improves the matching.
>
> 2) When CRM and DRM are used together, the individual performance gains relative to the baseline are partially offset.
> This suggests a degree of overlap in the sets of samples selected for re-matching by the two methods. To minimize redundant computation, we apply CRM after DRM in our pipeline.
>
> 3) The model's performance is not sensitive to the execution order of CRM and DRM, further probing their complementarity and necessity.
>
> | Method             | Sup-21K | iBOT-21K | iBOT-1K | DINO-1K | MoCo-1K |
> |--------------------|---------|----------|---------|---------|---------|
> | Baseline           | 71.52   | 73.60    | 72.98   | 70.43   | 67.98   |
> | Baseline+DRM       | 72.60   | 74.09    | 73.58   | 71.12   | 68.69   |
> | Baseline+CRM       | 72.80   | 74.35    | 74.01   | 71.32   | 68.80   |
> | Baseline+DRM+CRM   | 73.40   | 74.75    | 74.18   | 71.85   | 69.10   |
> | Baseline+CRM+DRM   | 73.61   | 75.10    | 74.22   | 71.80   | 69.18   |
>
> ---
>
> **Q6: The technical and typographical errors**
>
> **A6:** Thanks for the helpful comment. Line 193 states: For fairness, both our method and HiDe-Prompt use LoRA as the PET, and therefore HiDe-Prompt is referred to as HiDe-LoRA in the table. We promise to correct grammar, formulas, unclear expressions, and other technical and typographical errors in the revision.
>
> ---

---

> > ### Comment · Reviewer_b4of · 2025-08-02
> >
> > Thank you very much for your kind feedback. I deeply appreciate thoughtful feedback. However, I still have a few concerns about the following:
> >
> > 1.The proposed method appears to share its core concept with a previously published approach [1], which raises a concern about its novelty. Furthermore, since models for subsequent tasks are not trained on samples from previous tasks, it is unclear how they maintain valid prediction distributions for these past tasks. Prior works [1,2] address this by modeling feature distributions to synthesize prototypes. Could the authors clarify the specific mechanism in their method that mitigates this issue?
> >
> > [1] Sun, Hai-Long, et al. "Mos: Model surgery for pre-trained model-based class-incremental learning." *AAAI*. 2025.
> >
> > [2] Zhou, Da-Wei, et al. "Expandable subspace ensemble for pre-trained model-based class-incremental learning." *CVPR*. 2024.
> >
> > 2.The motivation behind the CRM design requires further explanation. My understanding is that its effectiveness relies on both $E(\hat{d}(x))$ and $E(\tilde{d}(x))$ being low. Is this interpretation correct? It is unclear how the method handles cases where one metric is high and the other is low. The design seems to focus primarily on $E(\tilde{d}(x))$, which could lead to incorrect mismatch identification when $E(\hat{d}(x))$ is high. Could the authors elaborate on this design choice and its justification?
> >
> > 3.Could the authors comment on the scalability of the proposed method? Specifically, does the inference time increase with the number of incremental tasks? It appears that the re-matching process for a growing number of past tasks could significantly increase computational costs.

---

> ### Author Response · Authors · 2025-08-02
>
> Thank you for your insightful and timely feedback.
>
> **Q1: Difference from MOS and the mechanism for maintaining valid prediction distributions for past tasks**
>
> **A1:**
>
> **Difference from MOS:**
> Although MOS [1] also addresses the problem of matching inaccuracy through Self-Refined Retrieval Mechanism,
> their strategies and assumptions are fundamentally different from ours:  MOS assumes that the sample is mismatched when the task identity $\hat{t}\_{s}$ of the final predicted class is not same as the task identity  $\hat{t}\_{f}$ corresponding to the selected PTE parameters. Specifically, MOS utilizes different task identities for inference until $\hat{t}\_{s} = \hat{t}\_{f}$. However, not all samples satisfy this assumption. In the incremental process, each task-specific PET's parameter and the final class classifier are trained jointly.  When an input is inferred by a PET parameter with an incorrect task ID, the final classifier is likely to activate classes belonging to that same (erroneous) task. Therefore, although $\hat{t}\_{f}$ and $\hat{t}\_{s}$ are the same, they may both have wrong task identities. For example, in all mismatched samples of ImageNet-R, 55.71\% of them are $\hat{t}\_{f} \neq \hat{t}\_{s}$. In contrast, our HRM-PET adopts a confidence-based rematching, under a different assumption for detecting and correcting mismatching based on prediction confidence rather than task identity consistency.
>
> **The mechanism for maintaining valid prediction distributions for past tasks:**
> We follow the advanced PET-based method HiDe-Prompt to maintain valid prediction distributions for past tasks by modeling the categorical features as a Gaussian distribution and storing them. We will add these details and cite [1] and [2] as related works in the revision.
>
> ---
>
> **Q2:The motivation behind the CRM.**
>
> **A2:** Thank you for your helpful comments. We clarify the motivation of CRM as follows:
>
> 1) CRM is divided into two stages, i.e., detecting mismatched samples and finding the correct task identity for the mismatched samples. During the detection phase, the sample with $E(\tilde{d}(x))$ meeting the threshold is determined as mismatching.
> This process only depends on whether the task identity used in inference is correct, regardless of the high or low of $E(\hat{d}(x))$.
>
> 2) During the finding the correct task identity, we obtain the top-N candidate task identities from $\hat{d}(x)$ based on the predicted probability, so $E(\hat{d}(x))$ potentially affects the recall of the correct task identity.
> On the one hand, we can increase N to increase the candidate range.
> On the other hand, in Equation 5, we compare the distribution confidence after inference with all candidate task identities to avoid selecting the wrong candidate identity.
>
> ---
>
> **Q3:Discussion of scalability**
>
> **A3:** To discuss the resource consumption on longer tasks, we conduct extra experiments on 20 tasks on ImageNet-R as shown in the table below. Since $N$ in CRM is set to 2, the inference cost of our method only depends on the number of mismatched samples, e.g., the number of samples that meet the confidence threshold. When the number of tasks increases to 20, the matching error rate is higher, which inevitably leads to an increase in inference time. **However, we improve the utilization of inference resources**. Our method achieves a higher gain in accuracy per millisecond of increased inference time. The inference time per image only increases by 0.57 ms, but the accuracy has increased by 3.1\%. Certainly, further reducing absolute inference cost remains a promising direction.
> | Method   | 10 Tasks        |               | 20 Tasks        |               |
> |----------|-----------------|---------------|-----------------|---------------|
> |          | $A_N \uparrow$ | Time (ms) $\downarrow$ | $A_N \uparrow$ | Time (ms) $\downarrow$ |
> | Baseline | 71.60           | 2.81          | 68.03           | 2.81          |
> | HRM-PET  | 73.86           | 3.24          | 71.13           | 3.38          |
>
> ---

---

> > ### Comment · Reviewer_b4of · 2025-08-04
> >
> > I thank the authors for answering my comments.
> >
> > I believe that the authors' responses can solve most of my concerns of this paper, and I would like to re-rate this paper to 4.

---

> > > ### Author Response · Authors · 2025-08-04
> > >
> > > Dear reviewer b4of,
> > >
> > > Thank you for kindly providing invaluable suggestions on this work. We authors greatly appreciate the efforts you have made to improve our manuscript. In the revised manuscript, we will add detailed discussions as suggested. If accepted, we will include `b4of` in our acknowledgments.
> > >
> > > Best Regards, Authors of paper 16355

---

### Official Review · Reviewer_F34S · 2025-06-28

**Clarity:** 2
**Significance:** 3
**Originality:** 3
**Rating:** 5
**Confidence:** 4

**Summary:**

This manuscript focused on the problem of continual learning with pre-trained models (Cl-PTM). Specifically, the authors focused the performance of task identity matching during the inference process, in which the task identity was predicted for each test sample, and then the task-specific parameters (e.g,  prompts) were selected to make the final classification. The authors investigated that the task-identity match between test samples and task-specific parameters was still challenging in the existing methods, and usually suffered from a high mismatching rate. To address this issue, the authors proposed HRM-PET, a parameter-efficient tuning based on a re-matching strategy, to improve the match accuracy in the existing methods. Specifically, the predicted distribution with the parameters selected through the initial matching was further used to improve the matching results, one by direct re-matching and another by confidence-based re-matching. Besides, the authors further introduced cross-task instance relationship distillation to better acquire the task-invariant knowledge.

**Questions:**

1. About the content between Lines 148–156: I find Section 3.2 clear until Eq. (5), but I struggle to grasp the intuition and derivation behind it. Could the authors elaborate on this equation in more detail? I have checked the supplementary materials but did not find a thorough explanation.
2. Missing training details: Some training details are lacking, which makes it difficult to evaluate the computational complexity. For example, in Section 3.3, a subset $ \mathbf{s}_i $ is selected for $ x_i $. However, the computation of this subset, as well as the subsequent steps in Eq. (8), seems to rely on $ e_k(x) $, i.e., inference using previous task parameters. Does this mean that for each new task $ t $, we need to compute the feature embedding $ e_k(x) $ for each sample under task $ t $ using all previous task-specific parameters $ p_k $? If so, this may introduce additional training overhead. Could the authors provide clarification and, if possible, quantitative evidence? It would also be helpful to include an algorithmic summary in the supplementary material.
3. Minor expression improvements: In the first line of Eq. (7), I suggest using $\hat{d}_{c_j}(x_i)$ instead of $\hat{d}_{c_j}$ to improve clarity.

Overall, I do not have major concerns about this work. I look forward to the authors’ clarifications, which will inform my evaluation during the rebuttal phase and further reviewer discussions.

**Ethical Concerns:**

["NO or VERY MINOR ethics concerns only"]

**Final Justification:**

I appreciate the effort from the authors during the rebuttal period. After carefully reading the responses from the authors and the reviews from my colleagues, I decided to raise my rating to "Accept".

**Quality:**

3

**Strengths And Weaknesses:**

## Strengths
1. This study investigated the core and long-existing problem in CL-PTM, the potentially inaccurate task-parameter matching process during the test phase, which is crucial in this area.
2. The motivation of the proposed method was clear and evidently supported by empirical observations (e.g., Figure 1-a).
3. The experimental part is relatively complete. Most of the recent baseline methods and the commonly used benchmarks are included. The ablation studies were clear and supportive. The results succeed to support the motivation of the proposed method.

## Weaknesses
1. One intrinsic trait is the proposed method is the need for extra inference compared to existing methods, only with initial matching. However, according to the empirical analysis, it seems that the extra inference process did not introduce too much computational overhead.
2. The expressions of some detailed steps are confusing. More clarification can be made to help the readers better understand the technical details. See the Questions part for more details.

---

> ### Author Rebuttal · Authors · 2025-07-30
>
> Thank you for your insightful comments and questions.
>
> **Q1: Clarification of inference time**
>
>
> **A1:** The minimal additional inference time overhead is mainly attributed to the following three factors:
> 1) Not all samples require additional inference in the re-matching. For DRM, we perform extra inference only on samples where $\hat{t}\_{f}$ is not equal to $\hat{t}_{s}$. For CRM, re-matching is conducted on samples whose confidence scores satisfy $\tau$. Hence, the average inference time per image increases marginally. In the table below, we provide the **Ratio** of samples undergoing re-matching under each strategy for clearer interpretation.
> | Method            | $A_{N}\uparrow$ | Time$\downarrow$ | Ratio |
> |------------------|-----------------|------------------|-------|
> | Baseline          | 71.60           | 2.81             | -     |
> | Baseline+DRM     | 72.60           | 2.91             | 12.2  |
> | Baseline+CRM+DRM |  73.86           | 3.24             | 30.3  |
> 2) In CRM, $N$ is set to 2, only one additional forward pass through the ViT is required.
> 3) In the baseline, each sample requires two forward passes through the ViT: one for task identity prediction and another for class prediction. Re-matching process involves only a single forward pass through the ViT, so the inference time does not double that of the baseline.
>
> ---
>
> **Q2: Explanation of Eq. (5)**
>
>
> **A2:** Eq.(5) describes how to find a more appropriate task identity when the sample $x$ is mismatched. Specifically, we first obtain the top-$N$ classes with the highest probabilities from the initial matching prediction distribution $\hat{d}(x)$:
>
>
> $
>     \text{$\Gamma$} = \underset{\{c\_j\}\_{j=1}^N}{\text{argmax}}\ \hat{d}\_{c_j}(x)
> $
>
> Where $\text{$\Gamma$}$ is the set of the top-$N$ classes. Candidate task identities are obtained by converting all classes to task identity in $\text{$\Gamma$}$ with $\mathcal{T}$. Then, we compare the confidence scores $E$ of the final prediction distributions generated by inference with the parameters $p$ corresponding to all task identities. The task identity with the highest confidence is selected as the corrected task identity $\hat{t}_{s}$:
>
> $
>     \hat{t}\_{s} = \underset{i \in \text{$\Gamma$}}{\mathrm{arg\,max}}\ E(g(h(x;p\_{\mathcal{T}(i)},\theta\_{ptm});\theta\_{g}))
> $
>
> The final class prediction ${\hat{y}}\_{CRM}$ is derived from the prediction distribution corresponding to $\hat{t}\_{s}$:
>
> $
>     {\hat{y}}\_{CRM} = \underset{i}{\text{argmax}} \, g(h(x;p\_{\hat{t}\_{s}},\theta\_{ptm});\theta\_{g})
> $
>
> ---
>
> **Q3: Missing training details**
>
>
> **A3:** In fact, for each new task $t$, we compute the feature embedding $ e_k(x) $ for each sample under task $t$ using the top-$K$ ($K=5$) previous task-specific parameters, rather than all old parameters. Specifically, we first utilize the pre-trained task classifier $g_{\omega}$ to obtain the top-$K$ old task identities with the highest confidence scores for all samples. The time cost of performing a forward through the classifier is negligible. Subsequently, before training, each sample undergoes $K$ inferences to obtain $K$ features for knowledge distillation. Compared to training, the inference-only forward introduces minimal additional time overhead.
>
> Experimentally, we measure the training time of baseline and HRM-PET on ImageNet-R on an RTX 3090 with a batch size of 64.  As shown in the table, our method incurs only an additional 4.4\% (i.e., 0.30 hours) of time overhead, which is acceptable. We will add the discussion in the revision.
>
> | Method        | Baseline    | HRM-PET (ours) |
> |---------------|-------------|----------------|
> | Training Time | 6.75 hours  | 7.05 hours     |
>
> ---
> **Q4: The expressions of some detailed steps are confusing**
>
>
> **A4:** Thanks for the helpful suggestion. We promise to fix the confusing expression in the revision.
>
> ---

---

> > ### Comment · Reviewer_F34S · 2025-08-04
> > **Author Rebuttal Follow-up**
> >
> > I appreciate the response from the authors. After reading it, my questions have been answered. Before the internal discussion period, I decided to maintain my current positive rating for this manuscript.

---

### Official Review · Reviewer_RzxM · 2025-07-02

**Clarity:** 3
**Significance:** 2
**Originality:** 3
**Rating:** 3
**Confidence:** 4

**Summary:**

This paper studies how to perform continual learning with a pretrained model using the Parameter-Efficient Tuning module. It proposes some techniques (direct rematching + confidence rematching) to improve the task identity prediction when selecting PET task module during inference. It also proposes a distillation loss for training to encourage the learn of shared knowledge among different task PET modules.

The paper presents an extensive evaluation of the proposed method, compared against several baseline methods, on four datasets and different pretrained models. It also presents ablation studies to examine the effectiveness of each algorithm component as well as key hyperparameters.

**Questions:**

1. The confidence is a key component of the proposed method and is used in both the rematching step and distillation step (e.g. Eq 3 and 7). To compute confidence, the paper mentioned the use of the generalized entropy function with additional hyperparameters $gamma$ and $M$. Why not use other hyperparameter-free confidence-related measures, like predicted probability or entropy? How are $gamma$ and $M$ selected? What is the interplay between generalized entropy function hyperparameter and the threshold parameter?

2. The paper mentioned that the proposed method can be applied to different PETs. Any results to support this claim?

3. As the proposed method involves many algorithm-specific hyperparameters, how are these algorithm-specific hyperparameters selected? What about baseline methods?

**Ethical Concerns:**

["NO or VERY MINOR ethics concerns only"]

**Final Justification:**

The paper is technically sound. I acknowledge the strengths of paper mentione by other reviewers. However, due to increased inference time, the limited theoretical depth, and the introduction of algorithm-specific hyperparameters, I maintain my initial rating of 3 at this stage.

**Limitations:**

yes, but a bit short.

**Paper Formatting Concerns:**

No.

**Quality:**

3

**Strengths And Weaknesses:**

Strengths
- The idea of direct rematching (section 3.2.1) is simple and effective in improving task identification prediction.
- The paper includes an extensive comparison with baseline methods on four datasets with different pretrain models.
- The paper provides ablation studies to investigate the effectiveness of the proposed three components,
- The paper analyzes the effect of some of the key hyperparameters.

Weaknesses
- Confidence plays an important role in the proposed method, which is used in both rematching and distillation. The authors mentioned the use of generalized entropy to compute confidence. But in the appendix, it seems that different confidence functions are used for different datasets (maxlogits for CIFAR100). It is unclear how to select the confidence measure for different datasets. It is better to include a systematic study on the effect of the confidence measure.

- The proposed method significantly increases inference time. Although the authors mentioned the absolute increase is small (0.4ms), the relative inference time is about 15% (from 2.8 to 3.2 ms). The inference time cost is already considered a big and important problem when employing foundation models in practice.

- The proposed method introduces many hyperparameters, e.g.$\gamma$, $M$, $\tau$, N, K, $\lambda$. How to select a suitable combination of these hyperparameters for a dataset is non-trivial.

- There is no error bar or confidence interval or standard deviation in the ablation studies, e.g. Table 3, 4 and Figure 6.

---

> ### Author Rebuttal · Authors · 2025-07-30
>
> Thank you for your insightful comments and questions.
>
> **Q1: A systematic study on the effect of the confidence measure**
>
>
> **A1:**
> To explore the effect of the confidence measure, we conduct ablation experiments on different post-hoc confidence measures in the table.
>
> 1) Overall, our method is not sensitive to different confidence calculation methods.
>
> 2) Generalized entropy (GEN) performs best across most settings.  This is attributed to its design for semantic shift scenarios, i.e., detecting inputs with semantic categories that are absent from the training set [48].  In CL, the PET parameters for task $t$ are only trained on the semantic categories of task $t$.  When predicted task identity $\hat{t}$ does not match the true task $t$ for sample $x$ from task $t$, the semantic class of $x$ is absent from the training set in task $\hat{t}$.  Therefore, GEN is more advantageous in detecting mismatched samples.
>
> 3) On CIFAR-100, performance is near the upper bound with few mismatched samples. MaxLogit with a high threshold effectively filters mismatches, so we adopt it for better performance.
>
> | Confidence | Sup-21K (CIFAR-100) | iBOT-21K (CIFAR-100) | Sup-21K (ImageNet-R) | iBOT-21K (ImageNet-R) | Sup-21K (ImageNet-A) | iBOT-21K (ImageNet-A) |
> |:----------:|:-------------------:|:--------------------:|:--------------------:|:---------------------:|:--------------------:|:---------------------:|
> | MSP        |        89.23        |        89.20         |        73.73         |         74.65         |        44.12         |         40.59         |
> | MaxLogit   |      **89.45**      |      **89.70**       |        73.77         |         74.97         |        44.20         |         40.60         |
> | Energy     |        89.24        |        89.29         |        73.60         |         75.12         |        44.14         |         40.73         |
> | GEN        |        89.35        |        89.55         |      **73.86**       |       **75.23**       |      **44.28**       |       **40.88**       |
>
> ---
>
> **Q2: Increasing inference time**
>
>
> **A2:** Although the inference time is relatively increased, we achieve state-of-the-art performance across multiple datasets and pretraining settings. We acknowledge its practical significance and investigate the accuracy-efficiency trade-off in Section 4 of the supplementary material. In the future, further improving efficiency remains a promising direction.
>
> ---
> **Q3&Q7: Selection of hyperparameters**
>
>
> **A3&A7:**
> Our method is robust to most hyperparameters, and their selection is generally straightforward:
> 1) $\gamma$ and $M$ are from generalized entropy (GEN).  According to GEN paper, we set $\gamma$ in (0, 1). $M$ represents the top-$M$ highest probability in the distribution. Given that the minimum number of classes encountered during the incremental process is 20 (in the second task) for all datasets, we set it to 20.
> 2) Given the inference cost, $N$ is set to 2 for all datasets.  As shown in Section 4 of the supplementary material, $N$ governs the trade-off between accuracy and efficiency.  $N$ can be adjusted based on available deployment resources.
> 3) $K$ and $\lambda$ in CTIRD are 5 and 0.2 for all datasets. We use common settings for $K$, i.e., top-5. In Eq. 8, $\mathcal{L}_{\mathrm{CTIRD}}$ is the sum of $K$ KL divergence losses in knowledge distillation. Drawing from prior work such as IRD [7], where the coefficient for a single distillation step is typically set to 1, we scale the overall loss by $1/K$, resulting in $\lambda = 1/K = 0.2$.  The experiment in Fig. 6 verifies the effectiveness of our choice.
> 4) The $\tau$ affects the accuracy of mismatch detection. We search for the optimal value on the validation set.  Overall, our method is robust to $\tau$.  As shown in Fig. 6 (c), treating all samples as mismatches with $\tau=10^3$ still yields an improvement of 0.77 compared to selecting none ($\tau=-10^3$).
> 5) For the baseline methods, we employ the same selection strategy with identical hyperparameters to ensure a fair comparison.
>
> ---
>
> **Q4:  The standard deviation in the ablation studies**
>
>
> **A4:** As shown in the three tables below, we supplement the standard deviations of three random seeds for some experiments in Tab. 3, 4, and Fig. 6 (a). The conclusions drawn remain consistent with those in the initial draft. We will include all standard deviations in the revision.
>
> ### Tab. 3
>
> | Method                     | Sup-21K           | iBOT-21K          |
> |:----------------------------:|:-------------------:|:-------------------:|
> | Baseline                   | 71.47 ± 0.26      | 73.49 ± 0.31      |
> | Baseline + DRM             | 72.55 ± 0.10      | 74.11 ± 0.22      |
> | Baseline + DRM + CRM       | 73.38 ± 0.21      | 74.80 ± 0.15      |
> | Baseline + DRM + CRM + CTIRD | **73.86 ± 0.14** | **75.23 ± 0.21**  |
>
>
> ### Tab. 4
>
> | Distillation | CIFAR-100 (Sup-21K) | (CIFAR-100 iBOT-21K) | ImageNet-R (Sup-21K) | ImageNet-R (iBOT-21K) |
> |--------------|-------------------|--------------------|--------------------|---------------------|
> | Logits       | 87.70 ± 0.14      | 88.10 ± 0.20       | 72.79 ± 0.28       | 74.32 ± 0.33        |
> | Features     | 87.29 ± 0.22      | 83.78 ± 0.18       | 71.19 ± 0.16       | 71.66 ± 0.13        |
> | IRD*         | 88.61 ± 0.15      | 88.86 ± 0.26       | 73.22 ± 0.18       | 74.68 ± 0.30        |
> | CTIRD        | **89.45 ± 0.23**  | **89.70 ± 0.13**   | **73.86 ± 0.14**   | **75.23 ± 0.21**    |
>
>
> ###  Fig. 6 (a)
>
> | PTM      | 0.0              | 0.1              | 0.2               | 0.3              |
> |----------|------------------|------------------|-------------------|------------------|
> | Sup-21K  | 73.38 ± 0.21     | 73.40 ± 0.19     | **73.86 ± 0.14**  | 73.18 ± 0.23     |
> | iBOT-21K | 74.80 ± 0.15     | 75.08 ± 0.20     | **75.23 ± 0.21**  | 74.96 ± 0.17     |
>
> ---
>
> **Q5: The hyperparameters of generalized entropy**
>
>
> **A5:** Considering advanced performance and advantages in detecting semantic drift, we choose GEN as the main confidence function. According to GEN paper, we set $\gamma$ in (0, 1). $M$ represents the top-M highest probability in the distribution. Given that the minimum number of classes encountered during the incremental process is 20 (in the second task), we set it to 20. GEN’s hyperparameters influence the range and distribution of confidence scores, which in turn affect the optimal threshold. By fixing $\gamma$ and $M$, we ensure consistency in the confidence scale, enabling a more stable threshold.
>
> ---
>
> **Q6: Results of different PETs**
>
>
> **A6:** As shown in the table, using prompt as PET, we show the $A_N$ on two different pre-trained weights on the ImageNet-R dataset compared to the state-of-the-art prompt-based method. Our method HRM-PET still achieves state-of-the-art performance.
> | Method         | iBOT-21K         | DINO-1K         |
> |----------------|------------------|-----------------|
> | CPrompt       | 64.64 ± 0.87     | 68.25 ± 1.43    |
> | HiDe-Prompt    | 70.83 ± 0.17     | 68.11 ± 0.18    |
> | HRM-PET (Prompt version) | **72.11 ± 0.19** | **69.55 ± 0.28**|
>
> ---

---

> > ### Comment · Reviewer_RzxM · 2025-08-04
> >
> > Thank you for your detailed rebuttal. The analysis of the confidence measures and hyperparameters is particularly reassuring.
> >
> > After revisiting the paper and considering the other reviewers' comments, I’d appreciate the authors' insights on the following:
> >
> > - ​​Blurry Class-Incremental Learning​​: Could the algorithm be adapted to a blurry class-incremental setting where class labels may overlap across tasks?
> >
> > - ​​Label Noise Sensitivity​​: How does label noise in the training data affect performance? Could the rematching scheme potentially amplify the impact of label noise?
> >
> > In addition, regarding the impact of the proposed method, it would be better to mention explicitly in the paper that there is a line of PET-based CL works (e.g. LAE, inferLoRA) that do not maintain task-specific parameters and thus do not require task identification in the first place.

---

> ### Author Response · Authors · 2025-08-05
>
> We sincerely appreciate your insightful comments and will add the following discussion to the revision.
>
> **Q1:​​Blurry Class-Incremental Learning​​**
>
> **A1:** Our algorithm can be applied to a blurry class-incremental learning scenario with some minor modifications:
> 1) In the blurry class-incremental learning scenario, some classes are associated with multiple task identities. However, in the initial task identity prediction of the baseline we follow, the model first predicts the class and then maps the class to a task identity with $\mathcal{T}$, which conflicts with blurry class-incremental learning setting.  To address this issue, we revise the initial task identity prediction by adopting the key-value mechanism from DualPrompt. Specifically, during training, a learnable key is maintained for each task, and the input image features are used as a query to retrieve the corresponding task identity through key-value matching.
>
> 2) Based on key-value mechanism, we make the following modifications to our algorithm: For DRM, we use the final predicted features as query to retrieve $\hat{t}\_{s}$, and then determine whether $\hat{t}\_{s}$ is equal to $\hat{t}\_{f}$.
> For CRM and CTIRD, key-value similarity is utilized to obtain the top-N or top-K task identities in Equation 5 and Equation 7.
>
> As shown in the table, we present a $A_N$ comparison of our method with the adapted version of HiDe-LoRA under blurry class-incremental learning scenario. We adopt the dataset split from [1] and iBOT-21K pretraining setting on ImageNet-R. * indicates that the key-value mechanism has been incorporated. As can be observed, our method also achieves the best performance.
>
> | Method   | HiDe-LoRA* | HRM-PET* (ours) |
> |----------|--------------|-----------------|
> | $A_N$    | 64.98        | **66.17**       |
>
> [1] Moon, Jun-Yeong, et al. "Online class incremental learning on stochastic blurry task boundary via mask and visual prompt tuning." Proceedings of the IEEE/CVF international conference on computer vision. 2023.
>
> ---
>
> **Q2:​​Label Noise Sensitivity**
>
> **A2:** As shown in the table below, we conduct experiments on Imagenet-R with iBOT-21K by injecting noise at different ratios. The label noise assigns {0\%, 10\%, 20\%} samples of the dataset to other labels by a uniform probability. Our method still achieves the best performance under different noise ratios. This benefits from our approach of comparing confidence scores in CRM to determine whether detected mismatched samples require task identity replacement, which alleviates the interference of noise on mismatching detection with threshold filtering. Moreover, Instance relation distillation in CTIRD is more robust to noise than logits or feature distillation, effectively learning valuable knowledge from previous tasks.
>
> | Method           | 0%    | 10%   | 20%   |
> |------------------|-------|-------|-------|
> | HiDe-LoRA      | 73.40 | 68.70 | 63.55 |
> | HRM-PET (ours)   | **75.23** | **70.13** | **64.81** |
>
> ---
>
> **Q3: Mention of PET-based CL works  that do not maintain task-specific parameters**
>
> **A3:** Thank you for your helpful suggestion. We will add the necessary details in the revision.
>
> ---

---

> > ### Author Response · Authors · 2025-08-06
> > **Looking Forward to Your Valuable Feedback**
> >
> > Dear reviewer,
> >
> > Thank you once again for your insightful and constructive feedback on our manuscript and rebuttal. We truly appreciate the time and effort you have dedicated to reviewing our work and engaging in the discussion.
> >
> > We have carefully considered each of your latest comments and have provided detailed responses. Following our discussion, reviewer `b4of` has raised the score from 2 to 4. We sincerely hope that our efforts also adequately address your concerns and contribute positively to your evaluation.
> >
> > Since the author-reviewer discussion period has been extended until Aug 8, we are fortunate to have sufficient time to collaboratively discuss and refine the manuscript. We would greatly appreciate any additional feedback you may have. If you have any additional questions or require any clarifications, please do not hesitate to reach out to us.

---

> > > ### Comment · Reviewer_RzxM · 2025-08-06
> > >
> > > Dear authors,
> > >
> > > Thank you for taking the time to address my questions and provide additional experimental results. I found the extension of your method to the blurry setting particularly interesting. The results under the noisy label setting also provide further evidence supporting the effectiveness of the proposed approach.
> > >
> > > Overall, I am convinced that the method is technically sound, and I have no further technical questions at this stage.
> > >
> > > However, due to the increased inference time, the introduction of several algorithm-specific hyperparameters, and the limited theoretical depth, I have decided to maintain my initial rating. That said, I acknowledge the strengths of the paper highlighted by other reviewers and would be comfortable with the paper going either way.

---

### Comment · Area_Chair_4E7V · 2025-08-01
**[ NeurIPS 25 ] reviewer-author discussions**

Dear Reviewers,

Thank you all for the big efforts. Please check authors' rebuttal to see if your original concerns have been addressed, as well as if you have any follow-up questions to the authors.

Dear Authors: Please engage with our Reviewers during this discussion period.

Thanks a lot.

---

### Note · Authors · 2025-08-13

Dear reviewers and AC:

We sincerely thank all reviewers for their time and valuable feedback. We appreciate that reviewers find our study investigates the core and long-existing problem (F34S), hybrid re-matching is innovative and effective (b4of, 4szz), experiments and ablations are comprehensive. (RzxM, F34S, b4of, 4szz).

In responses, we address the major concerns and additional comments from reviewers:
1) A study on confidence measures, discussions on inference time and hyperparameters, performance of different PETs and random seeds, extension to the blurry and the noisy label setting (RzxM).
2) Clarification of inference time, CRM, and training details (F34S).
3) An analysis of training time and re-matching, difference from previous approach, the mechanism for maintaining prediction distributions, the motivation of CRM, and scalability of the method (b4of).
4) Clarification of quality, originality, and details of figures, theory of re-matching, extension to reasonable framework (4szz).

**To summarize, we would like to re-emphasize the following key points:**

**Inference time:** We acknowledge a relative increase in inference time, but emphasize achieving SOTA performance with a marginal absolute increase. As discussed with Reviewer b4of, we also improve the utilization of inference resources on longer task sequences, showing scalability.

**Algorithm-specific hyperparameters:** While we introduce several hyperparameters, we have shown that our model is robust to most of them and their selection is straightforward and principled (A3&A7 with RzxM).

**The theoretical depth:** HRM-PET is simple yet effective and stable. We have stated our significant contributions (A2&A3 with 4szz). We provide further theoretical insight, casting re-matching within a reasonable probabilistic reasoning framework (A8 with 4szz), strengthening the theoretical foundation.

**Overall, we are trying to address all concerns and polish our paper in the revision:**
1. We will correct grammar, formulas, unclear expressions, and other technical and typographical errors.
2. We will add the studies on the confidence measure, different PETs, scalability, and extra ablation.
3. We will include the details of hyperparameters, training time, figures, PET-based CL methods, and differences from previous approach.
4. We will include the insight of study on blurry CIL and label noise setting, theory of re-matching improving performance and extension to more reasonable framework.

---

### Decision · Program_Chairs · 2025-09-17

**Decision:**

Accept (poster)

**Comment:**

This paper proposes Hybrid Re-matching with Parameter-Efficient Tuning (HRM-PET) for rehearsal-free continual class-incremental learning. The idea of integrating direct and confidence-based re-matching is technically sound and addresses an important challenge. Reviewers appreciated the clarity of the experiments, the practical utility of the approach, and the detailed rebuttal, which helped resolve most of initial questions.

At the same time, some concerns remain: increased inference time,  additional algorithmic hyperparameters, and the lack of deeper theoretical justification, as well as  the writing that could be further refined. While the rebuttal alleviated several issues, these points were not fully resolved.

Overall, this paper received 1 Accept, 2 Borderline Accept, and 1 Borderline Reject. The reviews were split: some upgraded to an acceptance recommendation after rebuttal, while others maintained reservations. Balancing the strengths and remaining weaknesses, this paper makes a meaningful empirical contribution to continual learning, though with limitations. This meta-review agrees on both the positive aspects and the negative aspects pointed out by the reviewers, and leans towards **Accept** this submission.  The authors are encouraged to carefully address all concerns in their revision.